# INSTRUCTRAG: INSTRUCTING RETRIEVAL AUGMENTED GENERATION VIA SELF-SYNTHESIZED RATIONALES

**Zhepei Wei    Wei-Lin Chen    Yu Meng**
Department of Computer Science
University of Virginia
{zhepei.wei,wlchen,yumeng5}@virginia.edu

## ABSTRACT

Retrieval-augmented generation (RAG) has shown promising potential to enhance the accuracy and factuality of language models (LMs). However, imperfect retrievers or noisy corpora can introduce misleading or even erroneous information to the retrieved contents, posing a significant challenge to the generation quality. Existing RAG methods typically address this challenge by directly predicting final answers despite potentially noisy inputs, resulting in an *implicit* denoising process that is difficult to interpret and verify. On the other hand, the acquisition of explicit denoising supervision is often costly, involving significant human efforts. In this work, we propose INSTRUCTRAG, where LMs *explicitly* learn the denoising process through self-synthesized rationales — First, we instruct the LM to explain how the ground-truth answer is derived from retrieved documents. Then, these rationales can be used either as demonstrations for in-context learning of explicit denoising or as supervised fine-tuning data to train the model. Compared to standard RAG approaches, INSTRUCTRAG requires no additional supervision, allows for easier verification of the predicted answers, and effectively improves generation accuracy. Experiments show INSTRUCTRAG consistently outperforms existing RAG methods in both training-free and trainable scenarios, achieving a relative improvement of 8.3% over the best baseline method on average across five knowledge-intensive benchmarks. Extensive analysis indicates that INSTRUCTRAG scales well with increased numbers of retrieved documents and consistently exhibits robust denoising ability even in out-of-domain datasets, demonstrating strong generalizability.[1]

## 1 INTRODUCTION

While large language models (LMs) have demonstrated remarkable text generation abilities (Brown et al., 2020; Team et al., 2023; Touvron et al., 2023), they may occasionally produce factually incorrect contents (Dhuliawala et al., 2023; Huang et al., 2023a; Ji et al., 2023; Sun et al., 2023; Xu et al., 2024d; Zhang et al., 2023), particularly when the task at hand requires the most current information or out-of-domain knowledge not adequately represented in the pre-training corpus (Jiang et al., 2023b; Shuster et al., 2021; Yu et al., 2023; Zhao et al., 2023). This limitation significantly hinders the reliable deployment of LMs in high-stakes domains where factuality is crucial (Magesh et al., 2024; Singhal et al., 2023; Xiao et al., 2021; Xiong et al., 2024).

In light of this, retrieval-augmented generation (RAG) (Asai et al., 2023b; Guu et al., 2020; Izacard et al., 2023; Khandelwal et al., 2019; Lewis et al., 2020) has been introduced to enhance the generation accuracy of LMs in knowledge-intensive tasks by leveraging the most up-to-date information and specialized knowledge from external sources (Kasai et al., 2024; Vu et al., 2023; Yang et al., 2024; Zhou et al., 2022). However, the retrieved contents are typically mixed with irrelevant or even erroneous information due to the absence of perfect retrieval solutions (Izacard et al., 2021; Karpukhin et al., 2020; Khattab et al., 2022; 2023; Shi et al., 2023; Su et al., 2024) and the presence

---

[1]Code is available at https://github.com/weizhepei/InstructRAG.

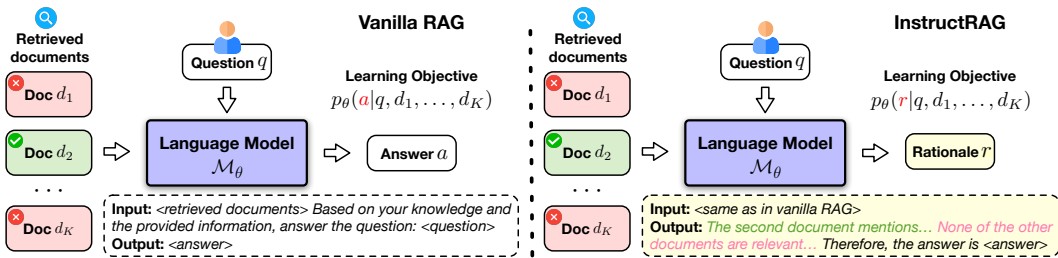

Figure 1: Comparison between vanilla RAG and our INSTRUCTRAG. In vanilla RAG, the model is tasked to directly predict answers given user queries and potentially noisy retrieved documents, without explicit denoising processes or explanations for how the answer is derived. In contrast, our proposed INSTRUCTRAG generates rationales that explicitly denoise the retrieved documents and justify the predicted answers, enhancing both the generation accuracy and trustworthiness.

of noisy data in the retrieval corpus (Izacard & Grave, 2021; Li et al., 2023; Yoran et al., 2024), posing a long-standing challenge to almost all RAG systems. Typically, vanilla RAG approaches address this issue *implicitly* by training LMs to directly predict correct answers despite noisy inputs. Such latent processes are not only difficult to interpret and verify but also vulnerable to higher noise ratios, especially when the number of retrieved documents is large (Chen et al., 2024; Cuconasu et al., 2024; Liu et al., 2024a; Wu et al., 2024). On the other hand, obtaining high-quality explicit denoising supervision often requires substantial human efforts, which is time-consuming and costly.

In this work, we introduce a new RAG framework, INSTRUCTRAG, which enables the LM to *explicitly* denoise retrieved information and justify its predicted final answers by generating denoising responses (*i.e.*, *rationales*), as illustrated in Figure 1. Compared to vanilla RAG approaches, INSTRUCTRAG does not require any additional supervision, while enjoying improved generation accuracy and trustworthiness. Specifically, our method consists of two steps. First, given a set of question-answer pairs and potentially noisy retrieved documents, we prompt an instruction-tuned LM to synthesize denoising rationales that analyze the documents and articulate how they lead to the ground-truth answers (§ 2.2). Then, these synthetic rationales can be utilized as in-context learning examples or as supervised fine-tuning data, allowing the LM to explicitly learn to denoise retrieved contents (§ 2.3). The effectiveness of INSTRUCTRAG can be attributed to the strong instruction-following ability of LMs (Jiang et al., 2024b; Ouyang et al., 2022; Wei et al., 2021), a significant feature that still remains underexplored in the context of RAG. We show that such self-synthesized rationales not only provide high-quality explicit denoising supervision for in-domain RAG tasks, but also facilitate superior out-of-domain generalization. This finding underscores how instruction-tuned LMs can synthesize generalizable supervision to overcome the inevitable noise in RAG.

The main contributions of this work are as follows: (1) We propose INSTRUCTRAG, a simple yet effective RAG framework that allows LMs to explicitly denoise retrieved contents by generating rationales for better verifiability and trustworthiness. (2) INSTRUCTRAG is a self-synthesis method that does not require additional supervision compared to standard RAG methods, and can be seamlessly applied to both in-context learning and supervised fine-tuning settings. (3) INSTRUCTRAG consistently outperforms state-of-the-art RAG approaches, yielding a relative improvement of 8.3% on average compared to the best baseline method across five knowledge-intensive benchmarks. Extensive analysis and ablation studies further confirm the superiority of self-synthesized denoising rationales, and demonstrate INSTRUCTRAG's robust denoising ability against increased noise ratios and strong task transferability in various training-free and trainable scenarios.

## 2 OUR METHOD: INSTRUCTRAG

In this section, we first introduce our problem setting (§ 2.1) and then present the proposed framework INSTRUCTRAG that enables LMs to explicitly denoise retrieved contents. As shown in Figure 2, our method consists of two steps. First, we prompt an instruction-tuned LM (*i.e.*, rationale generator $\mathcal{M}_\phi$) to synthesize rationales that provide denoising supervisions (§ 2.2). These rationales aim to explain how to derive the correct answer from potentially noisy retrieved documents for each training sample. Then, we guide the LM (*i.e.*, rationale learner $\mathcal{M}_\theta$) to learn explicit denoising by leveraging these rationales as either in-context learning demonstrations or as supervised fine-tuning

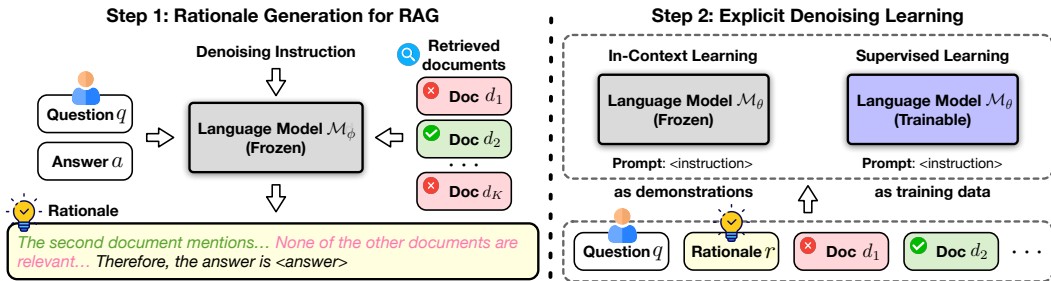

Figure 2: An overview of INSTRUCTRAG. In step one, given the question $q$, retrieved documents $\{d_1, \cdots, d_K\}$ and ground-truth answer $a$ from the training set, we prompt an instruction-tuned LM (*i.e.*, rationale generator $\mathcal{M}_\phi$) to generate rationale $r$ that explains how the answer can be derived from the potentially noisy input. In step two, we utilize the synthesized rationales from the first step to guide the LM (*i.e.*, rationale learner $\mathcal{M}_\theta$) to explicitly learn denoising of the retrieved documents, either through in-context learning or supervised learning. By default, we use the same model for both $\mathcal{M}_\phi$ and $\mathcal{M}_\theta$, but they can be instantiated with different models as well (see ablation study § 3.3).

Table 1: Rationale generation prompt for the $i$-th training sample.

| Rationale Generation |
| --- |
| **Input:** Read the following documents relevant to the given question: $\{q_i\}$ |
| Document [1] (Title: $\cdots$): $\{$contents of $d_i^1\}$ $\cdots$ Please identify documents that are useful to answer the given question: $\{q_i\}$, and explain how the contents lead to the answer: $\{a_i\}$ |
| $\{$task-specific instruction$\}$ |
| **Output:** $\{$rationale $r_i\}$ |

data (§ 2.3). As detailed in Algorithm 1, during the entire process, INSTRUCTRAG does not require any additional supervisions beyond standard RAG methods. By default, we instantiate both $\mathcal{M}_\phi$ and $\mathcal{M}_\theta$ with the same off-the-shelf instruction-tuned model (*i.e.*, meta-llama/Meta-Llama-3-8B-Instruct), making INSTRUCTRAG a fully *self-synthesis* method. We also experiment with different instantiations of $\mathcal{M}_\phi$ and $\mathcal{M}_\theta$ and conduct ablation study in both training-free and trainable settings (§ 3.3). For simplicity, we use placeholders to represent omitted instructions in the prompts presented in this section, while the full list of complete prompt templates is provided in Appendix D.

## 2.1 PROBLEM SETTING

We adopt the standard RAG setting where the LM $\mathcal{M}_\theta$ has access to annotated datasets of downstream tasks (*e.g.*, question-answering task $\mathcal{T} = \{\langle q, a \rangle\}$), and an external knowledge base with the off-the-shelf retriever $\mathcal{R}$ for retrieval. Different from previous works (Asai et al., 2023b; Yoran et al., 2024) which leverage additional supervisions from GPT-3 (Brown et al., 2020) or GPT-4 (Achiam et al., 2023), we assume the model has strictly limited access to the above two information sources. Given a question $q$, the retriever $\mathcal{R}$ returns a set of potentially noisy documents $D = \{d_1, \cdots, d_K\}$ from the external knowledge base. The model is then tasked to predict the correct answer $a$ to the given question $q$ based on $D$ and its own parametric knowledge, denoted as $p_\theta(a|q, D)$.

Our work focuses on investigating the noise robustness of LMs and developing efficient denoising techniques for RAG. Hence, we directly employ off-the-shelf retrievers instead of training our own, and prepend all retrieved documents to the question as input to the model, without any filtering or re-ranking. This setting is orthogonal to existing research efforts centered on optimizing the retriever or performing adaptive retrieval (Asai et al., 2023b; Wang et al., 2024a; Yang et al., 2024).

---

**Algorithm 1** INSTRUCTRAG

---

**Require:** Retriever $\mathcal{R}$, Rationale generator $\mathcal{M}_\phi$, Rationale learner $\mathcal{M}_\theta$, Training data $\mathcal{T} = \{\langle q, a \rangle\}$
    `/* Training data generation */`
1: **for** each $\langle q, a \rangle \in \mathcal{T}$ **do**
2:      Retrieve $D = \{d_1, \cdots, d_K\} \leftarrow \mathcal{R}(q)$
3:      Synthesize denoising rationale $r \leftarrow \mathcal{M}_\phi(q, a, D)$            ▷ Rationale Generation (§ 2.2)
4: Augment training data $\mathcal{T} \rightarrow \mathcal{T}^+ = \{\langle q, r \rangle\}$
    `/* Two learning modes */`
5: **if** LearningMode == In-Context Learning **then**       ▷ INSTRUCTRAG-ICL
6:      Sample ICL examples $\mathcal{E} = \{\langle q, r \rangle\} \subseteq \mathcal{T}^+$
7:      $r \leftarrow \mathcal{M}_\theta(r|q, \mathcal{R}(q), \mathcal{E})$ given inference query $q$       ▷ Detailed in Table 10
8: **else if** LearningMode == Fine-Tuning **then**       ▷ INSTRUCTRAG-FT
9:      Fine-tune $\mathcal{M}_\theta$ on $\mathcal{T}^+$ with retrieved documents $\{\langle q, r, D \rangle\}$
10:     $r \leftarrow \mathcal{M}_\theta(r|q, \mathcal{R}(q))$ given inference query $q$       ▷ Detailed in Table 11
11: **return** $r$

---

## 2.2 RATIONALE GENERATION VIA INSTRUCTION-FOLLOWING

Recent studies (Leike et al., 2018; Meng et al., 2024; Ouyang et al., 2022) have made encouraging progress in aligning LMs with human preferences and intentions, enabling the synthesis of high-quality data that closely follows user instructions (Xu et al., 2024c). Inspired by these advances, we propose to leverage the LM's strong instruction-following ability to generate explicit denoising responses (*i.e.*, *rationales*) for RAG. As shown in Table 1, given a QA pair $\langle q_i, a_i \rangle \in \mathcal{T}$ and a set of retrieved documents $\{d_i^1, \cdots, d_i^K\}$, we prompt an off-the-shelf LM $\mathcal{M}_\phi$ (as the rationale generator) with denoising instructions to produce the corresponding rationale $r_i$ that distinguishes useful documents from noisy ones and explains how the contexts lead to the ground-truth answer $a_i$. To ensure the synthetic rationales are aligned with the ground-truth answers, we use a simple substring match to assess their consistency. The consistency ratio on training samples with at least one relevant document containing the ground-truth answer is 98% on average across five benchmarks, supporting the reliability of synthetic rationales as a sanity check. This allows us to effectively augment the standard dataset $\mathcal{T} = \{\langle q, a \rangle\} \rightarrow \mathcal{T}^+ = \{\langle q, r \rangle\}$ with self-synthesized denoising rationales solely by instructing the LM, without any additional supervision.

We also validate the necessity of using an LM-based generator (*i.e.*, $\mathcal{M}_\phi$) to create the rationales instead of employing simple heuristics — without the generator, rationales can be created in a template-based manner (Table 6), by roughly identifying relevant retrieved documents through simple substring-matching with the ground-truth answer. However, as demonstrated in our ablation study, this approach suffers from semantically inaccurate matching of relevant documents, leading to significant performance degradation. Another advantage of the LM-based generator is that it can produce high-quality rationales even *without referring to the ground-truth answer*, which only results in a minor performance drop. More detailed analyses on rationale generation design can be found in our ablation study (§ 3.3).

## 2.3 LEARNING DENOISING RATIONALES IN RAG

With the rationale-augmented dataset $\mathcal{T}^+$, it becomes possible to develop a rationale learner $\mathcal{M}_\theta$ that directly learns explicit denoising for RAG tasks with efficient learning strategies. Next, we introduce two simple yet effective learning methods in the *training-free* and *trainable* RAG settings, namely, INSTRUCTRAG-ICL and INSTRUCTRAG-FT.

**INSTRUCTRAG-ICL** is a training-free instantiation of INSTRUCTRAG where the model learns denoising rationales via in-context learning (ICL). As shown in Table 10, given a test question $q$ and a set of retrieved documents $D = \{d_1, \cdots, d_K\}$, we first randomly sample $N$ demonstrations $\langle q_i, r_i \rangle \in \mathcal{T}^+$ from the rationale-augmented training dataset, and then prompt the model to follow the exemplars and generate rationale $r$. To save memory and enhance inference efficiency, we only show exemplary questions and their corresponding rationales in such ICL demonstrations.

**INSTRUCTRAG-FT** is a trainable instantiation of INSTRUCTRAG that learns denoising rationales via supervised fine-tuning (FT) with standard language modeling objective. As defined in Eq. 1, it maximizes the likelihood of rationale $r$ conditioned on question $q$ and retrieved documents $D$.

$$\max_{\theta} \mathbb{E}_{(q,r) \sim \mathcal{T}^{+}} \log p_{\theta}(r|q, D). \tag{1}$$

where $\theta$ represents the model parameters. Both the training and inference of INSTRUCTRAG-FT share the same data format. As depicted in Table 11, it takes as input the retrieved documents followed by the question, and outputs the denoising rationale $r$.

## 3 EXPERIMENTS

### 3.1 EXPERIMENTAL SETTING

Table 2: Dataset statistics and retrieval setting.

| Dataset | Train | Test | Retriever | Top-$K$ | Recall@$K$ |
|---|---|---|---|---|---|
| PopQA | 12,868 | 1,399 | Contriever | 5 | 68.7 |
| TriviaQA | 78,785 | 11,313 | Contriever | 5 | 73.5 |
| Natural Questions | 79,168 | 3,610 | DPR | 5 | 68.8 |
| ASQA | 4,353 | 948 | GTR | 5 | 82.2 |
| 2WikiMultiHopQA | 167,454 | 12,576 | BM25 | 10 | 40.7 |

**RAG tasks and evaluation metrics.** We extensively validate the effectiveness of INSTRUCTRAG on five knowledge-intensive benchmarks, including PopQA (Mallen et al., 2023), TriviaQA (Joshi et al., 2017), Natural Questions (Kwiatkowski et al., 2019), ASQA (Stelmakh et al., 2022), and 2WikiMultiHopQA (Ho et al., 2020). We use Wikipedia corpus as the retrieval source, and test our method with both sparse and dense off-the-shelf retrievers, including BM25 (Robertson & Walker, 1994), DPR (Karpukhin et al., 2020), GTR (Ni et al., 2022) and Contriver (Izacard et al., 2021). The retrieval quality is measured by Recall@$K$, indicating whether the retrieved $K$ documents contain the correct answer. Table 2 shows the detailed dataset statistics. Following standard evaluation settings (Asai et al., 2023b), we adopt the official metric of correctness (*str-em*), citation precision (*pre*) and recall (*rec*) for ASQA (Gao et al., 2023a), and use *accuracy* for the other tasks, which measures whether the ground-truth answers are included in the model generations (Mallen et al., 2023; Schick et al., 2023). Additionally, we also adopt LLM-as-a-judge for further evaluation (§ 3.4), as the above standard metrics are subject to the limitations of pattern-matching, which cannot accurately handle semantic equivalence.

**Baselines.** We compare our method with a wide range of RAG baselines under both training-free and trainable settings. Given that state-of-the-art LMs have incorporated a large amount of world-knowledge during the pre-training stage, we also report the performance of a non-retrieval baseline (namely, **vanilla zero-shot prompting**) for reference. Specifically, the training-free RAG baselines includes: (1) **in-context retrieval-augmented language modeling (RALM)** (Ram et al., 2023), a prompting method that extends the non-retrieval baseline by presenting the model with retrieved documents; (2) **few-shot demonstration with instruction**, an ICL method using ground-truth question-answer pairs sampled from the training set as demonstration exemplars.

The trainable RAG baselines include: (1) **vanilla supervised fine-tuning (SFT)**, a supervised method with the training objective of maximizing the data likelihood of ground-truth answer given potentially noisy input; (2) **RetRobust** (Yoran et al., 2024), which fine-tunes the RAG model on a mixture of relevant and irrelevant contexts to make it robust to irrelevant contexts; (3) **Self-RAG** (Asai et al., 2023b), a strong trainable baseline, focusing on adaptive retrieval controlled by special reflection tokens. Both RetRobust and Self-RAG were originally built on Llama-2 (Touvron et al., 2023) with additional supervisions. For example, RetRobust augments the training data for multi-hop reasoning tasks (*e.g.*, 2WikiMultiHopQA) by prompting GPT-3 to decompose the original query and generate intermediate subqueries, and Self-RAG requires GPT-4 to generate additional reflective tokens to augment training samples.

For a fair comparison, we re-implement the two methods on Llama-2$_{7B}$ and/or Llama-2$_{13B}$ with augmented training data released by their authors, and report their performance as the higher one

Table 3: Overall results of INSTRUCTRAG and baselines on five knowledge-intensive benchmarks in training-free and trainable RAG settings. We re-implement baselines and report their performance as the higher one between the original scores and our reproduced results. * indicates the results copied from Asai et al. (2023b) for reference. "–" indicates the results are not reported in the original paper or not applicable (*e.g.*, some methods cannot produce citations). The best performance is highlighted in **bold**.

| Method | PopQA (acc) | TriviaQA (acc) | NQ (acc) | MultiHopQA (acc) | (em) | ASQA (pre) | (rec) |
|---|---|---|---|---|---|---|---|
| *Baselines w/o Retrieval* | | | | | | | |
| **Vanilla Zero-shot Prompting** | | | | | | | |
| ChatGPT* | 29.3 | 74.3 | – | – | 35.3 | – | – |
| Llama-3-Instruct$_{8B}$ | 22.8 | 69.4 | 46.6 | 45.6 | 30.6 | – | – |
| Llama-3-Instruct$_{70B}$ | 28.9 | 80.6 | 57.9 | 57.5 | 39.1 | – | – |
| *RAG w/o Training* | | | | | | | |
| **In-Context RALM** (Ram et al., 2023) | | | | | | | |
| ChatGPT* | 50.8 | 65.7 | – | – | 40.7 | 65.1 | 76.6 |
| Llama-3-Instruct$_{8B}$ | 62.3 | 71.4 | 56.8 | 43.4 | 40.0 | 62.1 | 66.4 |
| Llama-3-Instruct$_{70B}$ | 63.8 | 76.3 | 60.2 | 51.2 | 43.1 | 62.9 | 67.6 |
| **Few-Shot Demo. w/ Instruction** | | | | | | | |
| Llama-3-Instruct$_{8B}$ | 63.1 | 74.2 | 60.1 | 45.3 | 42.6 | 55.0 | 64.4 |
| Llama-3-Instruct$_{70B}$ | 63.9 | 79.1 | 62.9 | 53.9 | 45.4 | 49.3 | 57.1 |
| **INSTRUCTRAG-ICL** | | | | | | | |
| Llama-3-Instruct$_{8B}$ | 64.2 | 76.8 | 62.1 | 50.4 | 44.7 | **70.9** | **74.1** |
| Llama-3-Instruct$_{70B}$ | **65.5** | **81.2** | **66.5** | **57.3** | **47.8** | 69.1 | 71.2 |
| *RAG w/ Training* | | | | | | | |
| **Vanilla Supervised Fine-tuning** | | | | | | | |
| Llama-3-Instruct$_{8B}$ | 61.0 | 73.9 | 56.6 | 56.1 | 43.8 | – | – |
| **Self-RAG** (Asai et al., 2023b) | | | | | | | |
| Llama-2$_{7B}$ | 55.8 | 68.9 | 42.4 | 35.9 | 30.0 | 66.9 | 67.8 |
| Llama-2$_{13B}$ | 56.3 | 70.4 | 46.4 | 36.0 | 31.4 | **70.3** | **71.3** |
| Llama-3-Instruct$_{8B}$ | 55.8 | 71.4 | 42.8 | 32.9 | 36.9 | 69.7 | 69.7 |
| **RetRobust** (Yoran et al., 2024) | | | | | | | |
| Llama-2$_{13B}$ | – | – | 39.6 | 51.5 | – | – | – |
| Llama-3-Instruct$_{8B}$ | 56.5 | 71.5 | 54.2 | 54.7 | 40.5 | – | – |
| **INSTRUCTRAG-FT** | | | | | | | |
| Llama-3-Instruct$_{8B}$ | **66.2** | **78.5** | **65.7** | **57.2** | **47.6** | 65.7 | 70.5 |

between the original scores and our reproduced results. As our method adopts instruction-tuned Llama-3 as the backbone model, we also train RetRobust and Self-RAG with Llama-3-Instruct$_{8B}$ and optimize their performance through extensive hyper-parameters search. More details on implementation, including training, inference, and prompt design are available in Appendix B and Appendix D. We also present some case studies in Appendix C.

## 3.2 MAIN RESULT

Table 3 shows the overall experimental results, providing a comprehensive comparison between our INSTRUCTRAG and baseline methods in both training-free and trainable RAG settings.

**Baselines without retrieval.** As shown in the first block, the basic instruction-tuned models (Llama-3-Instruct$_{8B}$ and Llama-3-Instruct$_{70B}$) already achieve notable performance across all five benchmarks, with the 70B model exhibiting a surprisingly competitive performance of 80.6% on the TriviaQA. This observation suggests that the required knowledge for these tasks mostly falls within the LM's parametric knowledge, probably due to what is known as *data contamination* (*i.e.*, the presence of test data of downstream tasks in the pre-training data of LMs) (Golchin & Surdeanu, 2023; Jacovi et al., 2023; Magar & Schwartz, 2022).

**RAG without training.** The second block shows the comparison among training-free RAG methods. In-context RALM and few-shot demonstration with instruction methods generally achieve higher performance than the non-retrieval baseline, highlighting the importance of retrieval for knowledge-intensive tasks. Encouragingly, our INSTRUCTRAG-ICL consistently outperforms all

Table 4: Ablation study on the impact of ground-truth answer, retrieved documents, and model size on rationale generation, and the use of demonstrations during model inference. The results of our default setting in INSTRUCTRAG are underlined.

| Method | Trainable RAG Setting | | Training-free RAG Setting | |
| --- | --- | --- | --- | --- |
| | PopQA | ASQA | PopQA | ASQA |
| *Rationale Generation Design* | | | | |
| with both | 66.2 | 47.6 | 64.2 | 44.7 |
| w/o ground-truth answer | 65.2 (↓ 1.5%) | 46.4 (↓ 2.5%) | 64.0 (↓ 0.3%) | 44.5 (↓ 0.4%) |
| w/o retrieved documents | 64.5 (↓ 2.6%) | 45.2 (↓ 5.0%) | 64.1 (↓ 0.2%) | 44.3 (↓ 0.9%) |
| *Model Size of Rationale Generator* | | | | |
| rationale template (no generator) | 59.6 (↓ 10.0%) | 46.3 (↓ 2.7%) | 60.0 (↓ 6.5%) | 41.4 (↓ 7.4%) |
| Llama-3-Instruct (8B) | 66.2 | 47.6 | 64.2 | 44.7 |
| Llama-3-Instruct (70B) | 67.0 (↑ 1.2%) | 49.1 (↑ 3.2%) | 64.8 (↑ 0.9%) | 47.9 (↑ 7.1%) |
| *Inference Strategy Comparison* | | | | |
| w/o demonstration | 66.2 | 47.6 | 63.0 (↓ 1.9%) | 43.1 (↓ 3.6%) |
| w/ demonstration | 66.1 (↓ 0.2%) | 44.7 (↓ 6.1%) | 64.2 | 44.7 |

training-free baselines across various metrics, confirming the effectiveness of self-synthesized denoising rationales. Moreover, the boost from 8B to 70B model indicates that INSTRUCTRAG-ICL scales effectively with larger backbone models, validating the generalizability of our method.

**RAG with training.** As present in the bottom block of Table 3, our INSTRUCTRAG-FT not only surpasses all non-retrieval and training-free baselines across all five benchmarks, but also significantly outperforms trainable RAG baselines on almost every metric. The only exception is in the ASQA task, where our method slightly underperforms Self-RAG in terms of citation (*i.e.*, *pre* and *rec*). This is because our work primarily focuses on explicit denoising for RAG to improve the correctness of generations, which is measured by *em*. Despite not being explicitly optimized for citation metrics, our method still achieves competitive citation performance, significantly enhancing both generation accuracy and trustworthiness. Note that RetRobust achieves competitive performance on 2WikiMultiHopQA, which involves multi-hop reasoning. We attribute this to the additional training supervision provided by GPT-3, which enables the model to explicitly generate intermediate subqueries and sub-answers. Another interesting finding is that Self-RAG consistently exhibits inferior performance compared to vanilla SFT, and even underperforms the training-free in-context RALM baseline across all benchmarks. We speculate the reason might be that these RAG tasks favor more domain-specific knowledge than general knowledge. However, it is challenging for Self-RAG to directly leverage in-domain features from existing training data as it requires GPT-4 to generate reflection tokens on these benchmarks, which is not available in our problem setting (§ 2.1).

## 3.3 ABLATION STUDY

**Providing ground-truth answers and retrieved documents is important for rationale generation.** As depicted in the first block of Table 4, we ablate the rationale generation design from two aspects: (1) *w/o ground-truth answer*, where the model has no access to the ground-truth answer during rational generation and must predict the answer and explain how it is derived solely based on retrieved documents; (2) *w/o retrieved documents*, where the model is not provided with any retrieved documents during rational generation, and in this case, it has to explain the given answer based on its own knowledge. Although it is not surprising that our default design consistently outperforms the two ablations, it is encouraging to find that our method still works well even without access to the retrieved documents or ground-truth answers. This finding suggests the great potential of our INSTRUCTRAG to operate in a fully unsupervised manner, which we believe is an exciting direction for future work.

**Larger rationale generator leads to better results.** The middle block shows how different sizes of rationale generators impact the performance of our method. It is evident that the template-based rationale generation method significantly underperforms our method, highlighting the necessity of rationale generator. This is because the template-based method relies on pattern matching to identify relevant documents containing the ground-truth answer, which only considers lexical similarity while ignoring semantic meaning. The neglect of semantics inevitably introduces noise in

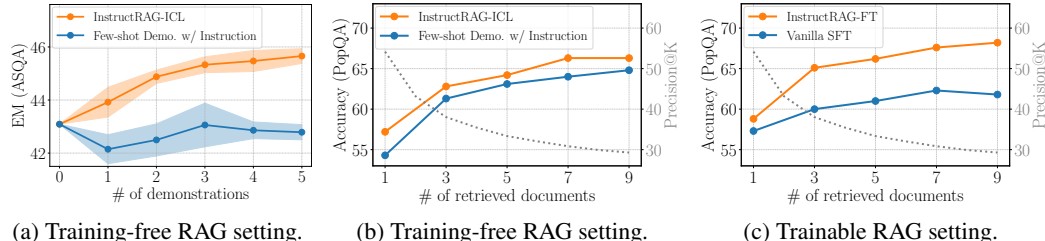

(a) Training-free RAG setting.     (b) Training-free RAG setting.     (c) Trainable RAG setting.

Figure 3: Impact of different number of demonstrations and retrieved documents. (a) Demonstration sensitivity study of INSTRUCTRAG-ICL. (b) Noise robustness study of INSTRUCTRAG-ICL. (c) Noise robustness study of INSTRUCTRAG-FT.

template-generated rationales, making them less effective compared to rationales generated by LMs. Moreover, we also compare two variants of INSTRUCTRAG using Llama-3-Instruct$_{8B}$ and Llama-3-Instruct$_{70B}$ as rationale generators. The results show that the one with a 70B generator consistently outperforms its 8B counterpart in both training-free and trainable settings, indicating that the self-synthesized denoising rationales can provide better supervision when generated by stronger models.

**Inference with demonstrations should only be applied to INSTRUCTRAG-ICL.** In the bottom block, we study the use of demonstrations during the model inference. While demonstrations play an important role for INSTRUCTRAG-ICL, we find that they actually hurt the performance of IN-STRUCTRAG-FT. We attribute this to the fact that INSTRUCTRAG-FT is optimized to directly generate denoising rationales given potentially noisy input, without referring to any demonstrations. Therefore, providing in-context demonstrations for INSTRUCTRAG-FT is redundant and may compromise its capability due to the discrepancy between training and inference.

## 3.4 ANALYSIS

**INSTRUCTRAG-ICL consistently benefits from more demonstrations.** Figure 3a shows the demonstration sensitivity of INSTRUCTRAG-ICL and the few-shot demonstration with instruction baseline. It is interesting to find that the baseline method achieves its best performance with only one demonstration, and presenting more demonstrations actually harms its performance. In contrast, our method consistently improves with the increasing number of demonstrations, confirming the superiority of self-synthesized rationales over plain answers in terms of denoising.

**INSTRUCTRAG-ICL and INSTRUCTRAG-FT are robust to increased noise ratios.** Figure 3b and Figure 3c show the generation accuracy of INSTRUCTRAG-ICL and INSTRUCTRAG-FT and the corresponding retrieval precision under an increasing number of retrieved documents. While retrieving more documents provides richer external knowledge to the RAG model, it also introduces more noise and lowers the retrieval precision. As a result, both the training-free and trainable baselines show diminishing improvements or even degrade as the number of documents increases, reflecting their vulnerability to high noisy ratios. In contrast, our INSTRUCTRAG-ICL and IN-STRUCTRAG-FT are not negatively affected by this increased noise ratio but rather gain further improvement, demonstrating their robust denoising ability.

**INSTRUCTRAG-ICL and INSTRUCTRAG-FT generalize well to unseen tasks.** Figure 4 demonstrates the generalization ability of our method in both training-free and trainable settings. For the in-domain (ID) method, it directly utilizes target domain demonstrations (in training-free settings) or is trained on the target domain task (in trainable settings). In contrast, the out-of-domain (OOD) method can only learn from demonstrations or training data in the source domain, and have no prior knowledge of the target domain. In this case, the model must leverage the knowledge learned from the source domain task to solve the unseen target domain task. The results show that our method consistently outperforms the baselines across various scenarios in both in-domain and out-of-domain settings, demonstrating strong task generalizability. One counter-intuitive finding is that in the scenario of generalizing from long-form to short-form QA task (Figure 4b), the training-free OOD method substantially outperforms its in-domain counterpart. We speculate that the training-free OOD method achieves better performance because it benefits from the demonstrations with long answers from the source domain (ASQA). The reason is that the questions in ASQA are ambiguous

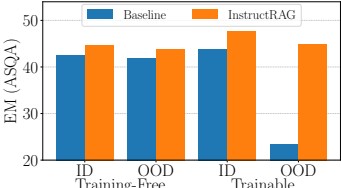 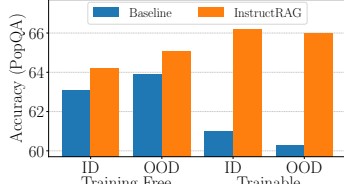 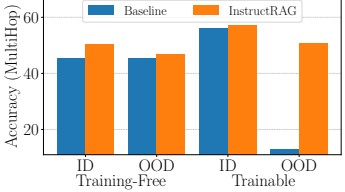

(a) Short-form to long-form QA.     (b) Long-form to short-form QA.     (c) Single-hop to multi-hop QA.

Figure 4: Generalizing INSTRUCTRAG from source domain task to target domain task, where ID and OOD denote in-domain and out-of-domain settings. (a) PopQA (short-form QA task) as source domain and ASQA (long-form QA task) as target domain. (b) ASQA as source domain and PopQA as target domain. (c) PopQA (single-hop QA task) as source domain and 2WikiMultiHopQA (multi-hop QA task) as target domain. We adopt *few-shot demonstration with instruction* and *vanilla supervised fine-tuning* as the training-free and trainable baselines.

Table 5: (a) Transfer from the source QA task (PopQA) to the target code generation task (HumanEval). Our method INSTRUCTRAG-FT is fine-tuned only on the source task and is evaluated on the unseen target task. We compare it with off-the-shelf LLaMA-3-8B-Instruct using the standard metrics pass@k (Chen et al., 2021), in both non-retrieval and retrieval-augmented generation settings. (b) Evaluation with GPT-4o as the judge. Compared to pattern-matching based metrics, it allows the judge to consider semantic equivalence and is expected to yield a more fair evaluation.

| Method | pass@1 | pass@10 |
|---|---|---|
| *Without Retrieval* | | |
| Llama-3-8B-Instruct | 58.5 | 64.6 |
| INSTRUCTRAG-FT | 60.4 | 65.2 |
| *With Retrieval* | | |
| Llama-3-8B-Instruct | 59.8 | 69.5 |
| INSTRUCTRAG-FT | 64.6 | 71.3 |

| Method | Pattern-based | LLM-based |
|---|---|---|
| *RAG w/o Training* | | |
| In-Context RALM | 56.8 | 64.5 |
| INSTRUCTRAG-ICL | 62.1 | 67.6 |
| *RAG w/ Training* | | |
| Vanilla SFT | 56.6 | 65.1 |
| INSTRUCTRAG-FT | 65.7 | 69.7 |

(a) Transfer from QA task to code generation task.     (b) Evaluation with GPT-4o as the judge.

and can have multiple interpretations, and ground-truth long answers often address the questions from various perspectives, which can be regarded as a form of chain-of-thought demonstration.

Furthermore, we also study the generalizability of INSTRUCTRAG to a non-QA knowledge-intensive task such as code generation. As presented in Table 5a, we directly apply INSTRUCTRAG-FT trained on the QA task (PopQA) to solve the unseen code generation task (HumanEval (Chen et al., 2021)), following the CodeRAG-Bench setup (Wang et al., 2024c). We evaluate the code generation performance using the standard pass@k metric and compare our method with the off-the-shelf Llama-3-8B-Instruct as the baseline. It can be observed that our method consistently achieves better generalization performance in the unseen code generation task in both non-retrieval and RAG settings. This finding aligns with our observation that INSTRUCTRAG trained on QA tasks tends to generate more text-based comments that articulate the design of coding solutions compared to the off-the-shelf Llama-3-8B-Instruct, thereby leading to more accurate code generation.

**Evaluation with LLM-as-a-judge.** Despite being standard evaluation metrics for question-answering, accuracy or exact match are known to be imperfect (Cuconasu et al., 2024) as they mainly rely on pattern-matching to judge the correctness of model predictions. Such metrics cannot handle cases where the prediction and ground-truth are synonyms (*e.g.*, "Donald Trump" vs "Donald J. Trump" cannot be correctly recognized as a match), leading to biased evaluation results. Therefore, we use LLM-as-a-judge (Bubeck et al., 2023; Zheng et al., 2024b) to evaluate the predictions with GPT-4o (OpenAI, 2024), which allows the judge to consider semantic equivalence and is expected to yield a more fair evaluation. As shown in Table 5b, we evaluate our method and baseline models on the open-domain Natural Questions benchmark in both training-free and trainable RAG settings. Compared to pattern-matching based metrics, LLM-as-a-judge generally leads to higher evaluation results, mostly due to its capability to accurately match semantically equivalent phrasings. Notably, our method consistently outperforms baselines under both pattern-matching based and LLM-based evaluation metrics, further validating the effectiveness of INSTRUCTRAG.

## 4 RELATED WORK

### 4.1 RETRIEVAL-AUGMENTED GENERATION

Retrieval-augmented generation (RAG) is a widely adopted approach to enhance large language models (LLMs) with external knowledge (Asai et al., 2023a; 2024; Borgeaud et al., 2022; Gao et al., 2023b; Guu et al., 2020; Khandelwal et al., 2019; Lewis et al., 2020; Ram et al., 2023; Shi et al., 2023), demonstrating promising potential to reduce hallucinations and enhance the generation accuracy of LLMs across various real-world applications (Chase, 2022; Jin et al., 2024b; Liu, 2022; Lu et al., 2022; Siriwardhana et al., 2023; Tan et al., 2024; Zhou et al., 2022; Liu et al., 2024b; Xiong et al., 2025). Recently, a growing research effort has been devoted to enhancing RAG from various aspects, such as improving decoding efficiency (Merth et al., 2024; Jin et al., 2024a; Liu et al., 2023; Wang et al., 2024b), exploring long-context retrieval (Xu et al., 2024a; Yen et al., 2024), compressing prompts (Jiang et al., 2023a; Xu et al., 2023; Cheng et al., 2024), and addressing practical concerns such as adversarial retrieval (Xiang et al., 2024; Zhong et al., 2023; Zou et al., 2024) and privacy leakage (Huang et al., 2023b; Zeng et al., 2024). Despite their advantages, these RAG systems inevitably suffer from irrelevant information introduced by imperfect retrievers or noisy retrieval corpora. However, most existing works typically address this issue by improving the retrieval quality and reducing noise exposure to the model (Gupta et al., 2024; Jiang et al., 2024a; Sarthi et al., 2024; Wang et al., 2024a; Yan et al., 2024; Yang et al., 2024; Zhang et al., 2024a;b). Notable methods include adaptive retrieval (Asai et al., 2023b; Jiang et al., 2023b; Yao et al., 2022) and query rewriting (Chan et al., 2024; Mao et al., 2024). In contrast, our work focuses on an orthogonal direction of developing explicit denoising methods for RAG, thereby enhancing the model's noise robustness and generation accuracy, even in highly noisy contexts.

### 4.2 ELICITING REASONING IN LARGE LANGUAGE MODELS

Recent studies have extensively explored the reasoning capability of LMs, but typically not in the context of RAG where potentially noisy retrieved contents may mislead the reasoning if not properly addressed. Chain-of-thought (CoT) prompting (Wei et al., 2022) is an effective method to elicit step-by-step reasoning from LMs by showing exemplars with detailed explanations (*i.e.*, rationales (Feng et al., 2024; Lampinen et al., 2022; Rajani et al., 2019; Zelikman et al., 2024)) that lead to the final answer. However, such works often requires manually crafted demonstrations (Wang et al., 2022; Xu et al., 2024b), which is costly and requires extensive efforts and domain knowledge (Zheng et al., 2024a). To mitigate this limitation, other methods have been introduced to automatically select instances from the corpus (Zhang et al., 2022) or curate demonstrations by the LM itself (Chen et al., 2023a), coupled with zero-shot CoT (Kojima et al., 2022) to generate rationales. Furthermore, it has been shown that CoT reasoning can be elicited even without explicit prompting, particularly for instruction-tuned LMs (Wang & Zhou, 2024). Another related work shows rationales generated by small models can help large models reason better (Lee et al., 2024). Although rationalization has been extensively investigated in many NLP tasks (Chen et al., 2023b; 2022; Ghoshal et al., 2022; Paranjape et al., 2020; Wiegreffe et al., 2021), none of them are designed for RAG, and how to leverage the instruction-following abilities of LMs for explicit denoising in the context of RAG still remains underexplored.

## 5 CONCLUSION

In this work, we presented INSTRUCTRAG, a simple retrieval-augmented generation (RAG) approach that explicitly denoises retrieved contents and produces accurate generations. By leveraging the strong instruction-following abilities of large language models, INSTRUCTRAG generates detailed rationales that articulate how the ground-truth answers can be derived from the retrieved documents. These synthetic rationales can serve as either in-context learning examples or supervised fine-tuning data, enabling the model to learn an explicit denoising process. Experiments on five knowledge-intensive benchmarks show INSTRUCTRAG consistently outperforms state-of-the-art RAG approaches with significant improvements in both training-free and trainable settings. Compared to the best baseline method, INSTRUCTRAG achieves an average improvement of 8.3% across all benchmarks, demonstrating its effectiveness in enhancing the noise robustness of retrieval-augmented generation. Limitations and future work are discussed in Appendix A.

## ACKNOWLEDGMENTS

The authors would like to thank Xinyu Zhu from University of Virginia, Tianyu Gao and Zexuan Zhong from Princeton NLP group for their valuable feedback and discussions. This research was supported in part by the NVIDIA Academic Grant and the OpenAI Researcher Access Program. We thank anonymous reviewers for their constructive and insightful comments.

## REPRODUCIBILITY STATEMENT

To ensure the highest level of reproducibility for our reported results, we have provided:

- Complete source code, accessible via the following link: `https://github.com/weizhepei/InstructRAG`;
- Comprehensive implementation details in Appendix B;
- All prompt templates used in our experiments in Appendix D.

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

## A   LIMITATIONS AND FUTURE WORKS

**Limitations.**   In this work, we mainly conduct experiments on question answering-type tasks, and it remains unclear how our method may generalize to other scenarios (*e.g.*, open-ended generation). Moreover, despite being the standard evaluation metrics, both accuracy and exact match are biased and cannot perfectly reflect the quality of the model's generations. For instance, such metrics heavily rely on string matching, which assesses correctness at the lexical level rather than the semantic level, thereby failing to recognize different phrasings that convey identical meanings. The evaluation results also suffer from length bias, as longer generations tend to achieve higher accuracy. Exploring more advanced metrics like using LLMs as judges would better evaluate RAG model generations (Yang et al., 2024). Another potential limitation is that our model might be subject to sample bias in the training data. Incorporating bias mitigation methods (Gallegos et al., 2024; Kawamae, 2023; Yang et al., 2023) would be helpful for further improving our work.

**Future Work.**   Future research directions include exploring more advanced techniques for generating high-quality rationales, such as incorporating domain-specific knowledge or leveraging multi-task learning to enable better generalization across various tasks. For instance, although the consistency ratio between synthetic rationales and ground-truth answers on training samples with at least one relevant document achieves 98%, the overall consistency ratio on all training samples is only 89%. This is because for some samples, none of the retrieved documents is relevant to the question, which significantly compromises the quality of the generated rationales. Therefore, it will be interesting to fully explore the potential of our method by incorporating additional designs such as a filtering mechanism, which we leave as future work. It will also be interesting to evaluate the model performance under long-context settings with a dynamic or extremely large number of retrieved documents. Finally, integrating our method with other advanced retrieval techniques (Su et al., 2024), such as active retrieval, could potentially lead to even better performance on knowledge-intensive tasks.

## B   IMPLEMENTATION DETAILS

**Retrieval setup.** Following (Asai et al., 2023b; Ram et al., 2023), we use the Wikipedia dump from (Karpukhin et al., 2020) as the external retrieval corpus for all five benchmarks studied in this work, where each document is a disjoint text block of up to 100 words extracted from a Wikipedia article. We compared all RAG methods under a diverse retrieval environment with various sparse and dense retrievers and number of retrieved documents. Specifically, we use Contriever-MS MARCO as the retriever for PopQA and TriviaQA, DPR for Natural Questions, GTR for ASQA, and BM25 for 2WikiMultiHopQA. By default, we retrieve the top 5 documents from the retrieval corpus for each query in all tasks except 2WikiMultiHopQA, where the top 10 documents are retrieved. We use the official weights for all dense retrievers and the implementation from Pyserini (Lin et al., 2021) for the sparse retriever BM25.

**Training details.** Our models are trained on 4 Nvidia H100 GPUs with 80GB memory via full-parameter fine-tuning. We use fully sharded data parallelism (FSDP) for distributed training, along with FlashAttention (Dao, 2023) and bf16 mixed precision training enabled for computation efficiency. By default, all models are trained using the Adam optimizer (Kingma & Ba, 2014) for 2 epochs, with a batch size of 128, a learning rate of 2.5e-5, and a cosine learning rate schedule with 3% warmup steps. For the trainable baseline vanilla SFT, we use a slightly different learning rate of 2e-5 based on our hyper-parameter search results. To fairly compare with Self-RAG and RetRobust, we re-implement them using Llama-3-Instruct-8B. We also optimize their performance through an extensive hyper-parameter search with learning rates in [8e-6, 1e-5, 2e-5] and training epochs in [1, 2, 3]. For Self-RAG, we use a learning rate of 1e-5 with a single training epoch. For RetRobust, we use a learning rate of 2e-5 with two training epochs. The only exception is the training for RetRobust on 2WikiMultiHopQA, where we train the model for 5 epochs on the augmented training set released by the original authors. The maximum token length for all models is fixed at 4096.

**Inference details.** By default, the number of demonstrations used in INSTRUCTRAG-ICL and the baseline method few-shot demonstration with instruction is set to be 2. We use vLLM (Kwon et al., 2023) to load models for memory-efficient inference and adopt the greedy decoding strategy for model generation.

# C CASE STUDY

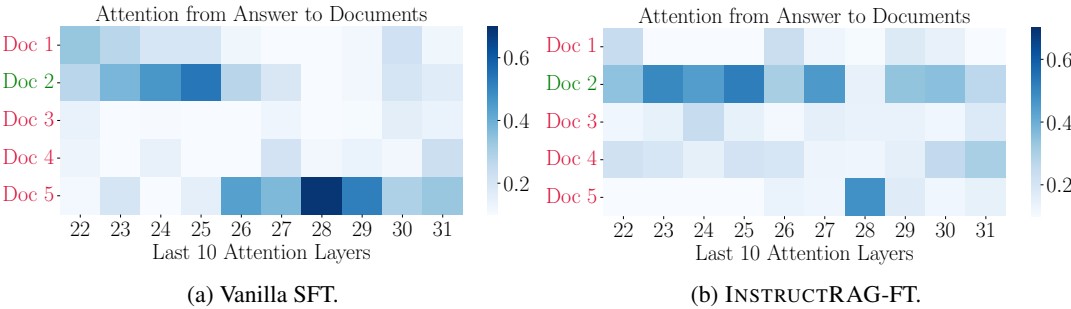

(a) Vanilla SFT.      (b) INSTRUCTRAG-FT.

Figure 5: Visualization of model attention from answer to retrieved documents on a random sample from the ASQA task, where Doc 2 is the only relevant document that contains the correct answer.

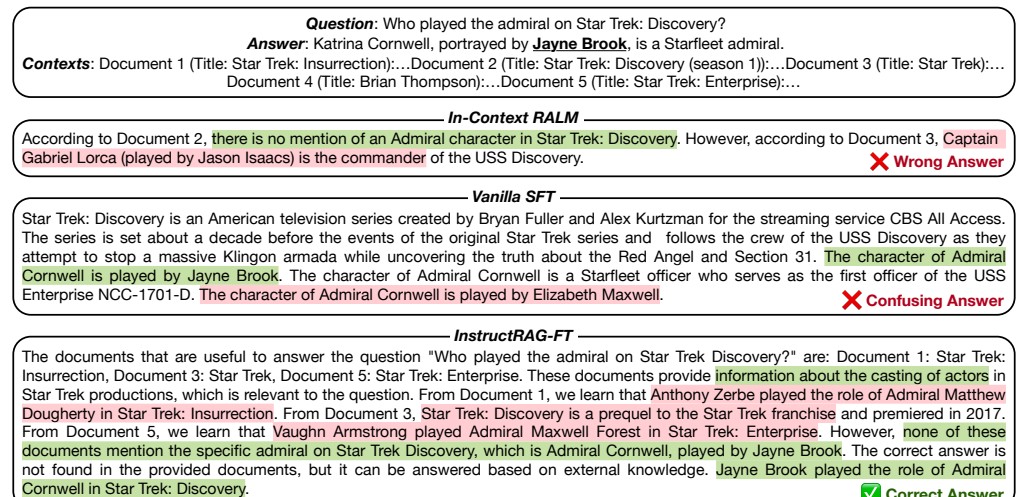

Figure 6: A case study of INSTRUCTRAG-FT compared with in-context RALM and vanilla SFT. The red texts denote irrelevant or inaccurate model generations, while the green texts denote contents relevant to the question. This study shows that our model can effectively identify relevant information from noisy input and leverage its own knowledge to correctly answer questions when required.

**Attention visualization.** To intuitively understand the denoising process of our INSTRUCTRAG, we visualize its attention from the answer to retrieved documents. As pointed out by a recent work (Yu et al., 2023), only attention distributions from deep layers can accurately reflect the LM's retrieval behavior and focus on key information, while attention from shallow layers usually do not imply meaningful patterns. Therefore, we only plot the attention weights of the last 10 layers (Layer 22 to Layer 31). As presented in Figure 5, our model accurately identifies the only benign document from noisy input, showing a strong denoising signal compared to vanilla SFT.

**Generation comparison.** Figure 6 compares the generated responses of in-context RALM, vanilla SFT, and our INSTRUCTRAG-FT for an actual question from the ASQA task. Among them, only our method can correctly answer this question while providing comprehensive denoising details. Specifically, it first identifies potentially relevant documents from noisy inputs, and then lays out the candidate information. More encouragingly, we find that INSTRUCTRAG-FT is able to refer to its own parametric knowledge when no relevant document is present in the context after denoising, demonstrating its superiority over existing RAG approaches.

## D PROMPT TEMPLATES

In this work, we instantiate the proposed INSTRUCTRAG with off-the-shelf instruction-tuned LMs (*i.e.*, meta-llama/Meta-Llama-3-8B-Instruct and meta-llama/Meta-Llama-3-70B-Instruct), and apply the official Meta-Llama-3-Instruct chat template (marked in gray) in all prompts.

**Rationale generation.** Below are the prompt templates for rationale generation used for all five benchmarks in our work. Table 6 shows the rationale template used in the ablation study (§ 3.3). For simplicity, we use the same prompt structure (Table 7) for all tasks with minor differences in task-specific instructions (Table 8).

Table 6: Rationale template used in ablation study.

| Rationale Template |
| --- |
| **Positive Template:** After reviewing the provided document, I found that only documents {documents} contain relevant information to answer the question. Based on my knowledge and the provided contents, the answer is: {answer}.

**Negative Template:** After reviewing the provided document, I found that none of them contain relevant information to answer the question. Based on my knowledge and the provided contents, the answer is: {answer}. |

Table 7: Rationale generation prompt template.

| Rationale Generation |
| --- |
| **Input:** `<|begin_of_text|><|start_header_id|>user<|end_header_id|>`
Read the following documents relevant to the given question: {question}

Document [1] (Title: · · · ): {contents}
· · ·
Please identify documents that are useful to answer the given question: "{question}", and explain how the contents lead to the answer: {answer}.

If none of the documents is aligned with the answer, in that case, you have to explain the answer only based on your own knowledge, without referring to the provided information.

**{task-specific instruction}**`<|eot_id|><|start_header_id|>assistant<|end_header_id|>`

**Output:** {rationale} |

Table 8: Task-specific instruction used in rationale generation prompt.

| Task-specific Instruction for Rationale Generation |
| --- |
| **ASQA:** Note that the question may be ambiguous and have multiple correct answers. Make sure your response includes all correct answers and provides clear reasoning details followed by a concise conclusion.

**PopQA:** Note that the question mainly asks about the object entity that holds a certain relationship with the given subject entity. There may be multiple correct answers. Make sure your response includes all correct answers and provides clear reasoning details followed by a concise conclusion.

**TriviaQA / Natural Questions / 2WikiMultiHopQA:** Note that the question may be compositional and require intermediate analysis to deduce the final answer. Make sure your response is grounded and provides clear reasoning details followed by a concise conclusion. |

**Inference prompts**. Below we present the inference prompts for both training-free and trainable RAG methods used in this work, including in-context RALM (Table 9), few-shot demonstrations with instruction (Table 10), and vanilla supervised fine-tuning (Table 11). Note that for a fair comparison, the inference prompt for our INSTRUCTRAG-FT is exactly the same as vanilla SFT. Similarly, the inference prompt for INSTRUCTRAG-ICL shares the same inference instruction as the few-shot demonstrations with instruction. The only difference between the prompts of these two methods lies in the demonstrations where INSTRUCTRAG-ICL employs denoising question-rationale $\langle q, r \rangle$ pairs, while few-shot demonstrations with instruction uses plain question-answer $\langle q, a \rangle$ pairs.

Table 9: Inference prompt for In-Context RALM.

---

**In-Context RALM**

**Input:** `<|begin_of_text|><|start_header_id|>user<|end_header_id|>`
Document [1] (Title: $\cdots$): {contents}
$\cdots$
Based on your knowledge and the provided information, answer the question: {question}
`<|eot_id|><|start_header_id|>assistant<|end_header_id|>`

**Output:** {answer}

---

Table 10: Inference prompt for INSTRUCTRAG-ICL and few-shot demonstrations with instruction.

---

**INSTRUCTRAG-ICL / Few-shot Demonstrations with instruction**

**Input:** `<|begin_of_text|><|start_header_id|>user<|end_header_id|>`
Your task is to analyze the provided documents and answer the given question. Please generate a brief explanation of how the contents of these documents lead to your answer. If the provided information is not helpful to answer the question, you only need to respond based on your own knowledge, without referring to the documents.

Below are some examples of how to answer the question:
{example question $q_1$}
{example answer $a_1$ / rationale $r_1$}
$\cdots$
Document [1] (Title: $\cdots$): {contents}
$\cdots$
Based on your knowledge and the provided information, answer the question: {question}
`<|eot_id|><|start_header_id|>assistant<|end_header_id|>`

**Output:** {answer}

---

Table 11: Inference prompt for INSTRUCTRAG-FT and vanilla supervised fine-tuning.

---

**INSTRUCTRAG-FT / Vanilla Supervised Fine-tuning**

**Input:** `<|begin_of_text|><|start_header_id|>user<|end_header_id|>`
Document [1] (Title: $\cdots$): {contents}
$\cdots$
Based on your knowledge and the provided information, answer the question: {question}
`<|eot_id|><|start_header_id|>assistant<|end_header_id|>`

**Output:** {answer}

---

## E  EXAMPLE OF LLM-AS-A-JUDGE

To measure the quality of model-generated rationales, we employ the LLM-as-a-judge approach for a more comprehensive evaluation of the model's outputs. As illustrated in Table 12, if the rationale is inaccurate despite the final answer being correct (probably due to the use of the LLM's parametric knowledge), the LLM judge will detect this inconsistency.

Table 12: Evaluate rationale with LLM-as-a-judge.

---

**Evaluation with LLM-as-a-judge**

**Input:** `<|begin_of_text|><|start_header_id|>user<|end_header_id|>`
You are a question-answering evaluator. Given the question, please judge whether the rationale aligns with the true answer.

{"question": "In what city was Catherine Tishem born?",
"true_answer": "Norwich, England",
"rationale": "The documents that are useful to answer the question are: Documents 4 and 5. Document 4 states that Catherine Tishem was from Breda, and her husband Wouter Gruter co-signed the Compromise of Nobles in 1566, and they moved to the Dutch Calvinist exile community of Norwich. This implies that Catherine Tishem was not born in Norwich. Document 5 provides more information about Catherine Tishem's life, stating that she was an erudite woman from Antwerp who educated her son, Jan Gruter, while in exile in England. This implies that Catherine Tishem was born in Antwerp. The other documents do not provide any information about Catherine Tishem's birthplace, so they are not useful for answering this question. Based on the above information, the answer is Norwich."}
`<|eot_id|><|start_header_id|>assistant<|end_header_id|>`

**Output:** The predicted answer concludes that Catherine Tishem was born in Norwich, which contradicts the conclusion drawn from the reasoning in the documents, where Antwerp is suggested as her birthplace based on the information in Document 5. Thus, while the predicted conclusion is Norwich, the rationale suggests Antwerp as the more likely birthplace based on the information in the documents. Therefore, the predicted answer does not align with the true answer.

---

