# OpenReview forum: "InstructRAG: Instructing Retrieval-Augmented Generation via Self-Synthesized Rationales"
_ICLR.cc/2025/Conference — ICLR 2025 Poster_

### Official Review · Reviewer_MKpj · 2024-10-25

**Soundness:** 4
**Presentation:** 4
**Contribution:** 3
**Rating:** 8
**Confidence:** 4

**Summary:**

The article introduces a simple yet effective RAG method to address noise problems in retrieved text results. Although the technique appears straightforward, I believe its simplicity enhances its versatility. I summarize the advantages as follows:

1. The approach is well-founded, as noise problems are prevalent in real-world RAG scenarios. InstructionRAG effectively addresses this problem.
2. The design is sound. The first phase generates rationales, which mitigate potential noise issues. Furthermore, the authors note, "The consistency ratio on training samples with at least one relevant document containing the ground-truth answer is 98% on average across five benchmarks, supporting the reliability of synthetic rationales as a sanity check." This aspect is crucial to the method's success.
3. The experiments are comprehensive, with validation across five datasets, demonstrating the effectiveness of InstructionRAG.

However, one might question whether using such sampling methods to address noise issues is worthwhile. This approach incurs additional time costs, which I believe cannot be overlooked. Exploring whether enhancing retrieval models might more effectively resolve this problem is a worthwhile consideration.

Overall, I am quite satisfied with the InstructionRAG method, which represents an influential contribution to the RAG field.

**Strengths:**

Please refer to the Summary section.

**Weaknesses:**

Please refer to the Summary section.

**Questions:**

Please refer to the Summary section.

---

> ### Author Response · Authors · 2024-11-22
> **Response to Reviewer MKpj**
>
> We sincerely appreciate the reviewer's positive feedback on our work and the constructive comments.
>
> **[Q1]**: However, one might question whether using such sampling methods to address noise issues is worthwhile. This approach incurs additional time costs, which I believe cannot be overlooked. Exploring whether enhancing retrieval models might more effectively resolve this problem is a worthwhile consideration.
>
> **[A1]**: Thanks for the suggestion. We agree that enhancing retrieval models is a valuable direction in RAG, which can reduce the noise at the retrieval stage. Nonetheless, the lack of perfect retrieves inevitably introduces some noise to RAG and thus this issue cannot be solely resolved by the retriever. Therefore, in our work, we address this by performing denoising during the generation stage, which is complementary to retrieval-stage denoising.
>
> We also acknowledge that our method might take slightly longer than standard RAG as it needs to generate more tokens (i.e., denoising rationales) before concluding the answer. Our experiments on the PopQA dataset show that InstructRAG incurs negligible latency compared to standard RAG (approximately 0.05 seconds slower per sample) but delivers a significant improvement in generation accuracy (61.0% → 66.2%).
>
> Please let us know if you have any further questions, and we are happy to incorporate additional suggestions that you might have! Thank you again for your time and support!

---

### Official Review · Reviewer_yPGP · 2024-10-29

**Soundness:** 3
**Presentation:** 3
**Contribution:** 3
**Rating:** 8
**Confidence:** 4

**Summary:**

This paper introduces InstructRAG, a model that learns denoising through self-generated rationales, which can be applied as in-context learning (ICL) demonstrations or supervised fine-tuning (SFT) training data to enhance performance. Experimental results across diverse datasets confirm the method’s effectiveness.

**Strengths:**

1. This paper proposes a RAG (Retrieval-Augmented Generation) paradigm that is suitable for both fine-tuning and in-context learning (ICL) and demonstrates its effectiveness. The design and validation of the entire approach are interesting.
2. This paper uses ablation experiments to prove that providing true answers and retrieved documents is very important for inference generation.
3. By comparing ICL and FT (fine-tuning) methods, the authors found that example-based reasoning should only be applied to InstructRAG-ICL, as it negatively impacts the performance of InstructRAG-FT. They provide reasoning for this, which is valuable for guiding future research.
4. Through out-of-distribution (OOD) experiments, the authors demonstrate that both InstructRAG-ICL and InstructRAG-FT can generalize well to unseen tasks.

**Weaknesses:**

1. If the complete answer to a question is distributed across multiple documents, with each document containing only part of the answer, this approach might mistakenly filter out these useful documents, leading to hallucinations or an inability to correctly answer the question. Could the authors provide more detailed analyses and experiments to clarify or verify the model's robustness?
2. The authors conducted experiments using only LLaMA-3-Instruct as the backbone, with the quality of self-synthesized rationales largely influenced by the underlying model's capability. Low-quality rationales may impair model performance, suggesting that the success of this approach could be attributed to LLaMA-3-Instruct's strong capabilities. Could the authors provide additional experiments with other backbones, such as LLaMA-2, Qwen, and Mistral, to help validate the method’s transferability and generalizability?
3. Although the datasets used in this paper include a variety of question-answering formats, such as short-form, multi-hop, and long-form, they are all based on wiki-style content. To more convincingly demonstrate the method's effectiveness, the authors should further extend their evaluation to more realistic and domain-diverse RAG datasets, like FreshQA and BRIGHT. Could the author also discuss the potential challenges in applying their method to these datasets and propose specific experiments to show the performance?
4. In Section 3.1, the experimental setting is unclear, do the authors utilize all the training samples for each dataset to generate rationales? Why do the authors adopt different retrievers for different datasets? What are the performance differences of the retrievers, such as Contriever, DPR, GTR, and BM25? Why is 2WikiMultiHopQA required to recall 10 documents while other datasets only require 5 documents?

**Questions:**

1. If the content in the documents does not explicitly contain the answer to the question, but the answer can still be inferred from the document content, can the rationale generator correctly reason and identify the relevant content? Did the authors conduct any corresponding ablation experiments or case studies to prove this?

---

> ### Author Response · Authors · 2024-11-22
> **Response to Reviewer yPGP [Part 1/2]**
>
> We sincerely thank the reviewer for the constructive suggestions, which have significantly improved our paper. We have completed all suggested experiments and added suggested analysis.
>
> **[Q1]**: If the complete answer to a question is distributed across multiple documents, with each document containing only part of the answer, this approach might mistakenly filter out these useful documents, leading to hallucinations or an inability to correctly answer the question. Could the authors provide more detailed analyses and experiments to clarify or verify the model's robustness?
>
> **[A1]**: We agree that this scenario presents an interesting challenge for RAG, which requires the model to accurately identify and combine relevant pieces of information. Indeed, our InstructRAG is especially effective for handling such cases by generating rationales that reason over multiple documents to deduce the final answer. Below is an illustrative example:
>
>
> > **Question**: Which country Audofleda's husband is from? \
> > **Ground-truth answer**: Ostrogothic Kingdom \
> > **Model Output**: After reviewing the provided documents, I found that none of them directly answer the question "Which country Audofleda's husband is from?". However, I can analyze the documents to deduce the answer. The documents that are useful to answer the question are:  Document 9 (Title: Audofleda),  Document 10 (Title: Audofleda), Document 4 (Title: Theoderic the Great). These documents provide information about Audofleda's marriage to Theoderic the Great, King of the Ostrogoths. Document 9 and 10 specifically state that Audofleda was the sister of Clovis I, King of the Franks, and that she married Theoderic the Great around 493 AD. Document 4 provides additional context about Theoderic's marriage to Audofleda, stating that it was a political move to ally himself with the Franks. The contents of these documents lead to the answer "Ostrogothic Kingdom" because they establish that Audofleda's husband, Theoderic the Great, was the King of the Ostrogoths. In conclusion, based on the information provided in the documents, Audofleda's husband is from the Ostrogothic Kingdom.
>
> This demonstrates our method’s ability to reason across multiple documents, accurately integrating distributed information to produce a coherent and correct answer.
>
>
> **[Q2]**: Could the authors provide additional experiments with other backbones, such as LLaMA-2, Qwen, and Mistral, to help validate the method’s transferability and generalizability?
>
> **[A2]**: Thanks for the suggestion. To validate the generalizability of our method, we further implemented InstructRAG with additional backbone models including Llama, Mistral, and Qwen. Below is the evaluation performance on the PopQA benchmark in both trainable and training-free settings.
>
> |Method|Llama3-8B-Instruct|Mistral-7B-Instruct-v0.3| Qwen2-7B-Instruct|
> |:-:|:-:|:-:|:-:|
> |||Training-free RAG||
> |In-Context RALM|62.3|60.8|60.1|
> |InstructRAG-ICL|64.2|62.2|62.5|
> |||Trainable RAG||
> |Vanilla SFT|61.0|60.0|60.3|
> |InstructRAG-FT|66.2|64.2|62.7|
>
>
> The above results further confirm the generalizability of our approach.
>
> **[Q3]**: Although the datasets used in this paper include a variety of question-answering formats, such as short-form, multi-hop, and long-form, they are all based on wiki-style content. To more convincingly demonstrate the method's effectiveness, the authors should further extend their evaluation to more realistic and domain-diverse RAG datasets, like FreshQA and BRIGHT. Could the author also discuss the potential challenges in applying their method to these datasets and propose specific experiments to show the performance?
>
> **[A3]**: We’d like to clarify that using wiki-style QA datasets is a standard practice and widely adopted in the RAG literature [1,2,3,4]. Nonetheless, we also validate our method on a non-QA knowledge-intensive task (**Code Generation**) to demonstrate its broader applicability. As presented in **Table 5(a)**, the results validate the generalizability of our method beyond wiki-style QA tasks.
>
> We appreciate the suggestion to explore more domain-diverse datasets, and we’ve added discussion on these works in our updated manuscript. While BRIGHT is a valuable benchmark for retrieval tasks, it does not provide direct answers for each query, making it challenging to evaluate question-answering methods. We have discussed BRIGHT in the introduction and future works of our updated manuscript.  To address the reviewer’s concern, we conducted additional experiments on FreshQA, evaluating our method on the most recently released data collection (2024-11-04):
>
> |Setting|Baseline|InstructRAG|
> |:-:|:-:|:-:|
> |Training-free RAG| 47.3 | 55.3|
> |Trainable RAG| 54.7 | 60.6|
>
> The results demonstrate that our InstructRAG consistently outperforms the baseline method in both training-free and trainable settings, further validating the effectiveness of our method in diverse scenarios.

---

> ### Author Response · Authors · 2024-11-22
> **Response to Reviewer yPGP [Part 2/2]**
>
> **[Q4]**: In Section 3.1, the experimental setting is unclear, do the authors utilize all the training samples for each dataset to generate rationales? Why do the authors adopt different retrievers for different datasets? What are the performance differences of the retrievers, such as Contriever, DPR, GTR, and BM25? Why is 2WikiMultiHopQA required to recall 10 documents while other datasets only require 5 documents?
>
> **[A4]**: Yes, the reviewer’s understanding is correct. As stated in Algorithm 1 (Lines 1-3), we generate rationales for all training samples in each dataset. We’d like to clarify that our method is agnostic to the choice of retrievers, and we directly follow the retrieval settings from prior work for each benchmark (e.g., Self-RAG[1] used Contriever for PopQA and TriviaQA, In-Context RALM[2] used DPR for NQ, ALCE[3] used GTR for ASQA, and FLARE[4] used BM25 for 2WikiMultiHopQA).
>
> As for the number of retrieved documents, 2WikiMultiHopQA requires retrieving 10 documents mainly due to its multi-hop reasoning nature. Since answering such questions involves integrating information from multiple sources, retrieving fewer documents would risk omitting crucial evidence, potentially compromising the RAG performance. Additionally, using different numbers of retrieved documents also provides a diverse test environment that allows us to compare different RAG methods with varying context lengths.
>
> **[Q5]**: If the content in the documents does not explicitly contain the answer to the question, but the answer can still be inferred from the document content, can the rationale generator correctly reason and identify the relevant content? Did the authors conduct any corresponding ablation experiments or case studies to prove this?
>
> **[A5]**: Thanks for the question. Yes, our InstructRAG is capable of performing reasoning over retrieved information to deduce the answer even when the answer is not explicitly stated in the retrieved documents. Below is an illustrative example:
>
> > **Question**: Who died first, Madame Pasca or James A. Donohoe? \
> > **Ground-truth answer**: Madame Pasca \
> > **Model Output**: The documents that are useful to answer the question "Who died first, Madame Pasca or James A. Donohoe?" are: Document 10 (Title: Madame Pasca), Document 9 (Title: Madame Pasca), Document 8 (Title: James A. Donohoe). The contents of these documents provide the necessary information to determine who died first. Document 10 and 9 provide the birth and death dates of Madame Pasca: November 16, 1833, and May 25, 1914, respectively. Document 8 provides the birth and death dates of James A. Donohoe: August 9, 1877, and February 26, 1956, respectively. By comparing the dates, we can conclude that Madame Pasca died first, on May 25, 1914, while James A. Donohoe died on February 26, 1956, more than 40 years later. Therefore, the answer to the question is: Madame Pasca.
>
>
> In this example, none of the documents explicitly state who died first but provide the necessary death dates. The model must correctly identify these dates and perform reasoning to answer the question. As demonstrated in the above model output, our method can effectively gather necessary information from multiple documents and deduce the correct answer.
>
> **Reference**
> - [1] Asai et al. Self-RAG: Learning to retrieve, generate, and critique through self-reflection. ICLR 2024.
> - [2] Ram et al. In-Context Retrieval-Augmented Language Models. TACL 2023.
> - [3] Gao et al. Enabling Large Language Models to Generate Text with Citations. EMNLP 2023
> - [4] Jiang et al. Active Retrieval Augmented Generation. EMNLP 2023.
>
> Please let us know if you have any further questions, and we are happy to incorporate additional suggestions you might have! If you find our response satisfactory, we would be grateful if you could consider raising your score. Thanks again for your time and efforts!

---

> > ### Comment · Reviewer_yPGP · 2024-11-29
> > **Thanks for your rebuttal**
> >
> > This work is both straightforward and insightful, and I greatly appreciate the authors’ efforts in providing a comprehensive response. As most of my concerns have been effectively addressed, I am pleased to raise my score to 8.
> >
> > Regarding the response to Q2, I acknowledge that models like Mistral-7B-Instruct-v0.3 and Qwen2-7B-Instruct are relatively recent. My original intention was to suggest that the authors include experiments on earlier versions, such as Qwen-7B-Chat or Mistral-7B-Instruct-v0.1, to demonstrate the robustness of the proposed approach. While I am not suggesting additional experiments at this stage, I hope the final version will include results on these earlier backbones to provide further insights.

---

> > > ### Author Response · Authors · 2024-12-02
> > > **Thanks to Reviewer yPGP for the acknowledgment**
> > >
> > > We are glad that our response addressed the reviewer’s concerns. Thank you again for the support and thoughtful comments, we will update our final version accordingly.
> > >
> > > Please let us know if you have any further questions, and we are happy to incorporate additional suggestions you might have!

---

### Official Review · Reviewer_GKWS · 2024-11-01

**Soundness:** 3
**Presentation:** 3
**Contribution:** 1
**Rating:** 5
**Confidence:** 4

**Summary:**

The paper introduces an instructed RAG method designed to explicitly learn the denoising process in language models using self-synthesized rationales. InstructRAG addresses these issues by instructing LMs to generate explanations on how ground-truth answers are derived from retrieved documents. These explanations are rationales utilized in two ways: in-context learning to teach explicit denoising without additional data and as data for supervised fine-tuning. This approach does not require extra human-labeled supervision, simplifies the verification of predictions, and improves generation accuracy.

**Strengths:**

1. The method effectively leverages existing data, reducing the need for costly new annotations, which often involve extensive human effort.

2. Demonstrating consistent outperformance over traditional RAG methods across multiple benchmarks indicates effectiveness.

**Weaknesses:**

1. The paper introduces a method that may involve noise in extracted knowledge or filtering information. However, the extent and nature of this noise are not adequately discussed. A quantitative measurement of the noise and its impact on the model’s performance would provide valuable insights into the robustness of the proposed method. Analyzing and quantifying the noise would also help clarify how much of the model’s learning is genuinely informative versus potentially spurious.

2. The paper would benefit from a more in-depth qualitative analysis of the extracted knowledge, including illustrative examples and rationales. Conducting a human study to evaluate whether the extracted knowledge aligns with useful, meaningful content (or if it simply reinforces dataset biases) would add a layer of rigor. This evaluation could help establish whether the proposed method provides actionable insights or merely reflects the underlying biases in the dataset.

3. The proposed method’s restriction on accessing additional information sources (GPTs), relying primarily on self-regularization, may limit its practical utility in real-world applications where additional context and supervision are often available. Furthermore, the filtering and re-ranking mechanisms can be viewed as forms of self-regularization, and evaluating the effects of incorporating or comparing these elements would provide a more balanced assessment. Testing the method with and without these components could illustrate its practical utility in more realistic scenarios.

4. The paper’s approach, which employs instruction tuning, in-context learning, and knowledge extraction, aligns closely with established methodologies in related fields. As a result, the method may lack significant novelty, as it primarily adapts a classical pipeline to the Retrieval-Augmented Generation (RAG) domain. Introducing new techniques or unique modifications tailored to RAG-specific challenges could significantly enhance the paper’s contribution to the field.

5. The paper would be strengthened by testing the proposed rationale learner across various models to assess its robustness. Demonstrating consistent performance improvements with different model architectures would reinforce the generalizability of the method and provide a more comprehensive evaluation of its effectiveness.

**Questions:**

A detailed analysis of the noise present in the dataset would be valuable for better understanding the robustness of the proposed method. More analysis of examples and rationales will be helpful. Testing it across different base models would be beneficial in assessing the robustness and generalizability of the proposed method.

---

> ### Author Response · Authors · 2024-11-22
> **Response to Reviewer GKWS [Part 1/3]**
>
> We sincerely thank the reviewer for the constructive suggestions, which have significantly improved our paper. We have completed all suggested experiments and updated our manuscript. Below we provide itemized responses to address the reviewer’s concerns.
>
> **[Q1]**: The extent and nature of noise are not adequately discussed. A quantitative measurement of the noise and its impact on the model’s performance would provide valuable insights into the robustness of the proposed method. Analyzing and quantifying the noise would also help clarify how much of the model’s learning is genuinely informative versus potentially spurious.
>
> **[A1]**: Below we present both qualitative and quantitative analysis to demonstrate that the retrieved documents can be highly relevant to the question but fail to provide precise information to answer the question, due to imperfect retrievers or potentially noisy retrieval corpus.
>
> **Qualitative analysis**: The following is a sample with noisy retrieval, where none of the retrieved passages mention the correct answer. For demonstration purposes, we only present the contents of the first two documents.
>
> > **Question**: Who got the first nobel prize in physics? \
> > **Ground-truth answer**: Wilhelm Conrad Röntgen \
> > Document [1] (title: Nobel Prize in Physics): The Nobel Prize in Physics is a yearly award given by the Royal Swedish Academy of Sciences for those who have made the most outstanding contributions for mankind in the field of physics. It is one of the five Nobel Prizes established by the will of Alfred Nobel in 1895 and awarded since 1901; the others being the Nobel Prize in Chemistry, Nobel Prize in Literature, Nobel Peace Prize, and Nobel Prize in Physiology or Medicine. \
> > Document [2] (title: Nobel Prize): A group including 42 Swedish writers, artists, and literary critics protested against this decision, having expected Leo Tolstoy to be awarded. Some, including Burton Feldman, have criticised this prize because they consider Prudhomme a mediocre poet. Feldman's explanation is that most of the Academy members preferred Victorian literature and thus selected a Victorian poet. The first Physiology or Medicine Prize went to the German physiologist and microbiologist Emil von Behring. During the 1890s, von Behring developed an antitoxin to treat diphtheria, which until then was causing thousands of deaths each year. The first Nobel Peace Prize went to the Swiss. \
> > Document [3] … \
> > Document [4] … \
> > Document [5] …
>
> **Quantitative analysis**: In **Section 3.1**, we also quantify the extent of noise across 5 benchmarks using recall@k, which indicates whether the top-k retrieved documents contain the correct answer. As presented in **Table 2**, the results show that a substantial portion of the samples lack the correct answer in their retrieved documents.
>
> Moreover, we also tested our model under different scenarios with varying noise ratios in **Section 3.4**. While retrieving more documents provides richer external knowledge to the RAG model, it also introduces more noise and lowers the retrieval precision. As demonstrated in **Figures 3(b) and 3(c)**, our method is not negatively affected by this increased noise ratio but rather gains further improvement in both training-free and trainable settings, demonstrating the robustness of InstrucRAG.

---

> ### Author Response · Authors · 2024-11-22
> **Response to Reviewer GKWS [Part 2/3]**
>
> **[Q2]**: The paper would benefit from a more in-depth qualitative analysis of the extracted knowledge, including illustrative examples and rationales. Conducting a human study to evaluate whether the extracted knowledge aligns with useful, meaningful content (or if it simply reinforces dataset biases) would add a layer of rigor. This evaluation could help establish whether the proposed method provides actionable insights or merely reflects the underlying biases in the dataset.
>
> **[A2]**: **Qualitative study**: Please refer to **Figure 6** for the suggested qualitative analysis, where we compare our method with the baseline method with a detailed example and rationales. This study shows that our model can effectively identify relevant information from noisy input and leverage its own knowledge to correctly answer questions when necessary.
>
>
> **Quantitative study**: In **Section 3.4**, we also utilize LLM-as-a-judge to quantitatively evaluate the rationales. This allows us not only to assess the correctness of the final answer but also to verify the validity of intermediate rationales: if the model-generated rationale is inaccurate despite the final answer being correct (probably due to the use of the LLM's parametric knowledge), the LLM judge still considers the prediction as incorrect – please refer to **Appendix E** in our updated manuscript for a detailed example. We report the LLM-based evaluation results in **Table 5(b)**, which demonstrate the reliability of the generated rationales.
>
> **Human Study**: Per the reviewer’s suggestion, we conducted a small-scale human study on 50 test samples where the model made a correct final prediction. In this study, we manually examined the rationales and found that 96% of the samples contained rationales that provided meaningful denoising contents and were consistent with the final answer. For the remaining 4% samples, none of their retrieved documents mention the correct answer, and the model only relied on its own knowledge to make the correct prediction. This human study further confirms the validity of the rationales.
>
> **[Q3]**: The proposed method’s restriction on accessing additional information sources (GPTs), relying primarily on self-regularization, may limit its practical utility in real-world applications where additional context and supervision are often available. Furthermore, the filtering and re-ranking mechanisms can be viewed as forms of self-regularization, and evaluating the effects of incorporating or comparing these elements would provide a more balanced assessment. Testing the method with and without these components could illustrate its practical utility in more realistic scenarios.
>
> **[A3]**: We would like to clarify that our method could indeed benefit from external supervision from stronger models. In **section 3.3 (Line 367)**, our experiments demonstrate that InstructRAG achieves further gains when external supervision is available to generate rationale guidance rather than relying solely on self-synthesized rationales. For instance, using Llama 3-70B to generate *external* supervising rationales for the Llama 3-8B model improves performance compared to using the 8B model alone, as shown in **Table 4**.
>
> Additionally, following the reviewer’s suggestion, we also incorporate a comparison study with filtering and re-ranking mechanisms as self-regularization and evaluate their performance on the PopQA dataset.
>
> |Model Size|Vanilla RAG| Vanilla + Re-ranking | Vanilla + Filtering| Ours|
> |:-:|:-:|:-:|:-:|:-:|
> |8B|62.3| 62.1|61.8|64.2|
> |70B|63.9|64.2|64.4|65.5|
>
> The results show that these methods do not reliably lead to improvements as the effectiveness of such self-regularization is highly dependent on a large model size. Moreover, their performance generally lags behind our method, which further demonstrates the superiority of InstructRAG.

---

> ### Author Response · Authors · 2024-11-22
> **Response to Reviewer GKWS [Part 3/3]**
>
> **[Q4]**: The paper’s approach, which employs instruction tuning, in-context learning, and knowledge extraction, aligns closely with established methodologies in related fields. As a result, the method may lack significant novelty, as it primarily adapts a classical pipeline to the Retrieval-Augmented Generation (RAG) domain. Introducing new techniques or unique modifications tailored to RAG-specific challenges could significantly enhance the paper’s contribution to the field.
>
> **[A4]**: We would like to clarify that the focus of our paper is denoising retrieved information using entirely self-synthesized rationales for RAG tasks. We summarize our key novelties as follows:
>
> - We use rationales for a novel purpose -- denoising retrieved passages, which is a unique challenge in RAG, whereas prior studies typically use rationales for reasoning instead of denoising [1,2].
> - We introduce a novel method for synthesizing rationales -- our self-synthesis approach using LLMs alleviates the requirements for human-annotated rationales [3,4].
> - We investigate the unique potential of instruction-tuned LLMs for RAG and demonstrate that they can teach themselves to generate rationales for explicit denoising, resulting in enhanced performance and better verifiability.
>
> **[Q5]**: The paper would be strengthened by testing the proposed rationale learner across various models to assess its robustness. Demonstrating consistent performance improvements with different model architectures would reinforce the generalizability of the method and provide a more comprehensive evaluation of its effectiveness.
>
> **[A5]**: Thanks for the suggestion. To validate the generalizability of our method, we further implemented InstructRAG with additional models including Llama, Mistral, and Qwen. Below is the evaluation performance on the PopQA benchmark in both trainable and training-free settings:
>
> |Method|Llama3-8B-Instruct|Mistral-7B-Instruct-v0.3| Qwen2-7B-Instruct|
> |:-:|:-:|:-:|:-:|
> |||Training-free RAG||
> |In-Context RALM|62.3|60.8|60.1|
> |InstructRAG-ICL|64.2|62.2|62.5|
> |||Trainable RAG||
> |Vanilla SFT|61.0|60.0|60.3|
> |InstructRAG-FT|66.2|64.2|62.7|
>
>
> The above results further confirm the generalizability of our approach.
>
> **Reference**
> - [1] Wang et al. Rationale-augmented ensembles in language models. arXiv:2207.00747
> - [2] Zelikman et al. STaR: Bootstrapping reasoning with reasoning. NeurIPS 2022.
> - [3] Yao et al. React: Synergizing reasoning and acting in language models. ICLR 2023.
> - [4] Wang et al. Self-consistency improves chain of thought reasoning in language models. ICLR 2023.
>
> Please let us know if you have any further questions, and we are happy to incorporate additional suggestions you might have! If you find our response satisfactory, we would be grateful if you could consider raising your score. Thanks again for your time and efforts!

---

> > ### Comment · Reviewer_GKWS · 2024-11-28
> >
> > I appreciate the authors' thoughtful reply, which addressed my concerns regarding the method's performance and robustness. In light of this, I have decided to raise my score. Thank you for your efforts in addressing these points and enhancing the clarity of the paper.

---

> > > ### Author Response · Authors · 2024-12-02
> > > **Thanks to Reviewer GKWS for the acknowledgment**
> > >
> > > Thank you for acknowledging our efforts to address the previous concerns and for raising the score! We appreciate your constructive feedback.
> > >
> > > If there are any remaining concerns that we haven't fully addressed, we would be grateful to know about them so we can further improve the paper. Thanks again for your time and efforts!

---

### Official Review · Reviewer_caq8 · 2024-11-04

**Soundness:** 3
**Presentation:** 4
**Contribution:** 3
**Rating:** 6
**Confidence:** 4

**Summary:**

This paper introduces INSTRUCTRAG, a novel approach to enhance retrieval-augmented generation by explicitly teaching language models to denoise retrieved information. Unlike traditional RAG methods that implicitly handle noisy inputs, INSTRUCTRAG leverages the instruction-following capabilities of LMs to generate synthetic rationales explaining how the correct answer is derived from the retrieved documents. These synthetic rationales serve as explicit denoising supervision, which can be used for in-context learning or supervised fine-tuning. This method not only improves the quality of in-domain RAG tasks but also enhances out-of-domain generalization. Finally, the paper demonstrates that by utilizing self-synthesized rationales, INSTRUCTRAG effectively addresses the challenges posed by noisy retrievals, making it a promising advancement in the field of RAG.

**Strengths:**

The paper effectively highlights the importance of denoising in RAG systems. This is a significant contribution to the field, as it addresses a critical issue that can significantly impact the performance of such models. The authors provide a thorough set of experiments that convincingly demonstrate the effectiveness of their proposed method. The results show that INSTRUCTRAG consistently outperforms both training-free baselines and training baselines across various metrics, which strongly supports the claim that self-synthesized denoising rationales are effective. Besides, the paper is well-organized and clearly written, making it easy for readers to follow the methodology, experimental setup, and results. The figures and tables are well-designed and enhance the understanding of the findings.

**Weaknesses:**

The paper could benefit from a more detailed description of the document retrieval process. Specifically, it is unclear whether the document collections used in the experiments contain noise samples, and if so, what the concentration of these noise samples is. This information is crucial for understanding the robustness of the proposed method and its applicability to real-world scenarios where noise data is often present. In addition, this paper could be strengthened by including a comparative analysis with other denoising techniques that have been proposed in the literature. This would help to contextualize the performance of INSTRUCTRAG and highlight its unique contributions.

**Questions:**

The paper presents an interesting approach to generating rationales for document retrieval tasks. However, there are several aspects that require further clarification and improvement to strengthen the validity and robustness of the proposed method. Below are my detailed comments:
1. It is unclear whether the document collections used in the experiments contain noise samples, and if so, what the concentration of these noise samples is. Understanding the presence and extent of noise is crucial for evaluating the robustness of the proposed method.
2. Beyond validating the effectiveness of the rationales through final experimental outcomes, how was the intrinsic quality of the generated rationales assessed? What measures were taken to ensure that the model-generated rationales are free from noise and that the selected documents are indeed highly relevant? This is important to confirm that the rationales are not only effective but also reliable and interpretable.

---

> ### Author Response · Authors · 2024-11-22
> **Response to Reviewer caq8 [Part 1/2]**
>
> We appreciate the reviewer's positive feedback and constructive comments.
>
> **[Q1]**: The paper could benefit from a more detailed description of the document retrieval process. Specifically, it is unclear whether the document collections used in the experiments contain noise samples, and if so, what the concentration of these noise samples is?
>
>
> **[A1]**: **Detailed description of the document retrieval process**: In this work, we use off-the-shelf retrievers to retrieve documents from the external Wikipedia dump. Due to the page limit, we provide the detailed retrieval configurations in **Appendix B**.
>
> **Example of retrieval noise**: Regarding the noise samples, we would like to note that the retrieved documents in RAG can be highly relevant to the question but fail to provide precise information to answer the question correctly due to imperfect retrievers or potentially noisy retrieval corpus. To illustrate this phenomenon more intuitively, the following is a sample with noisy retrieval from the test set, where none of the retrieved passages mention the correct answer. For demonstration purposes, we only present the contents of the first two documents.
>
>
> > **Question**: Who got the first nobel prize in physics? \
> > **Ground-truth answer**: Wilhelm Conrad Röntgen \
> > Document [1] (title: Nobel Prize in Physics): The Nobel Prize in Physics is a yearly award given by the Royal Swedish Academy of Sciences for those who have made the most outstanding contributions for mankind in the field of physics. It is one of the five Nobel Prizes established by the will of Alfred Nobel in 1895 and awarded since 1901; the others being the Nobel Prize in Chemistry, Nobel Prize in Literature, Nobel Peace Prize, and Nobel Prize in Physiology or Medicine. \
> > Document [2] (title: Nobel Prize): A group including 42 Swedish writers, artists, and literary critics protested against this decision, having expected Leo Tolstoy to be awarded. Some, including Burton Feldman, have criticised this prize because they consider Prudhomme a mediocre poet. Feldman's explanation is that most of the Academy members preferred Victorian literature and thus selected a Victorian poet. The first Physiology or Medicine Prize went to the German physiologist and microbiologist Emil von Behring. During the 1890s, von Behring developed an antitoxin to treat diphtheria, which until then was causing thousands of deaths each year. The first Nobel Peace Prize went to the Swiss. \
> > Document [3] … \
> > Document [4] … \
> > Document [5] …
>
> This example demonstrates that the retrieved documents can be highly relevant to the question but may not contain precise information about the answer, thereby increasing noise in the contexts.
>
> **Quantifying the noise ratio**: In **Section 3.1**, we quantify the noise percentage and measure retrieval quality using recall@k, which indicates whether the top-k retrieved documents contain the correct answer. As presented in **Table 2**, the results show that a substantial portion of the samples lack the correct answer in their retrieved documents. On the other hand, while retrieving more documents provides richer external knowledge to the RAG model, it also introduces more noise and lowers the retrieval precision. For example, as depicted in **Figures 3(b) and 3(c)**, the retrieval precision on the PopQA dataset is less than 60% when retrieving top-5 documents. These results confirm that the document collections used in the experiments indeed contain a considerable portion of noise samples.
>
> **[Q2]**: In addition, this paper could be strengthened by including a comparative analysis with other denoising techniques that have been proposed in the literature.
>
>
> **[A2]**: Thanks for the suggestion. Below is the comparative analysis of our method against a representative denoising method Llmlingua [1], which filters irrelevant content by compressing retrieved passages to retain only essential information. We tested various compression strategies on the PopQA benchmark, including dynamic compression rates, fixed compression (max compressed size = 200 tokens), and no compression. For a fair comparison, we employ both Llama-3-8B-Instruct and Llama-3-70B-Instruct as backbone models.
>
> |Model Size|No Compression|Fixed Compression|Rate=0.1|0.3|0.5|0.7|0.9| Ours|
> |:-:|:-:|:-:|:-:|:-:|:-:|:-:|:-:|:-:|
> |8B|62.3|48.2|31.7|43.9|54.2|61.1|61.8|64.2|
> |70B|63.9|50.3|37.7|51.3|60.5|64.3|64.4|65.5|
>
> The results show that our method consistently outperforms the existing denoising method in various scenarios, demonstrating the effectiveness of InstructRAG.

---

> ### Author Response · Authors · 2024-11-22
> **Response to Reviewer caq8 [Part 2/2]**
>
> **[Q3]**: Beyond validating the effectiveness of the rationales through final experimental outcomes, how was the intrinsic quality of the generated rationales assessed? What measures were taken to ensure that the model-generated rationales are free from noise and that the selected documents are indeed highly relevant?
>
>
> **[A3]**: We agree that it’s crucial to ensure the high quality of synthetic rationales. In light of this, we use two approaches to validate the quality of the generated rationales:
>
> - **Substring match**:  As introduced in **Section 2.2 (Line 185)**, we use a substring matching approach to assess the consistency between generated rationales and the ground-truth answers on training samples. It turns out that the consistency ratio can reach 98% on average across five benchmarks, demonstrating the reliability of such synthetic rationales.
>
> - **LLM-as-a-judge**: In **Section 3.4 (Line 465)**, we also measure the quality of model-generated rationales using LLM-as-a-judge, which allows for a more comprehensive evaluation of the model output. Specifically, if the model-generated rationale is inaccurate despite the final answer being correct (probably due to the use of the LLM's parametric knowledge), the LLM judge will detect this inconsistency – please refer to **Appendix E** in our updated manuscript for a detailed example. \
> In light of this, we further evaluate our method using GPT-4o as the judge. As shown in **Table 5(b)**, our method achieves even better performance under LLM based evaluation compared to pattern-matching based evaluation, which further confirms the reliability of the generated rationales.
>
>
> **Reference**
>
> [1]. Jiang et al. Llmlingua: Compressing prompts for accelerated inference of large language models. EMNLP 2023.
>
> Please let us know if you have any further questions, and we are happy to provide additional clarifications that could further enhance our work. If our response satisfactorily addressed your concerns, we would be grateful if you could consider raising your score. Thanks again for your time and efforts!

---

> ### Author Response · Authors · 2024-12-02
> **A Gentle Reminder to Reviewer caq8**
>
> Dear reviewer caq8,
>
>
> We sincerely appreciate your valuable feedback. We would like to gently remind you that the discussion period will **end on December 2nd**.  In our rebuttal, we have provided comprehensive responses to your comments, and we hope our rebuttal satisfactorily addressed your concerns.
>
> Please let us know if you have any further questions, and we are happy to incorporate additional suggestions you might have. Thank you again for your time and efforts in reviewing our paper!

---

### Official Review · Reviewer_PDdA · 2024-11-04

**Soundness:** 3
**Presentation:** 3
**Contribution:** 2
**Rating:** 8
**Confidence:** 3

**Summary:**

InstructRAG introduct one interesting method to explicitly learn the denoising process through self-synthesized rationales.  InstructRAG requires no additional supervision,  allows for easier verification of the predicted answers, and effectively improves generation accuracy.

**Strengths:**

1. InstructRAG is a simple but effective method to denoise retrvied information and justify its predicted final answers by generating denoising reponse
2. InstructRAG does not require any additional supervision and can be applied to both in-context learning and supervised fine-tuning settings
3. good presentation and  detailed experiment

**Weaknesses:**

1. The rationale data generatioin is crutial in InstructRAG.  I hope to get more details about how to denoise wrong rationales and how to ensure the generated rationales satisfy the corresponding standard.
2. I have some questions about how to use rationale in evaluation. From what I understand, InstructRAG needs to generate rationales first,  then concats the retrvied information and generated rationales to LM.  Maybe the added computation cost need to be analysized in the paper.

**Questions:**

The questions can be viewed in the weakness

---

> ### Author Response · Authors · 2024-11-22
> **Response to Reviewer PDdA**
>
> We appreciate the reviewer's positive feedback on our work and the constructive comments.
>
> **[Q1]**: The rationale data generation is crucial in InstructRAG. I hope to get more details about how to denoise wrong rationales and how to ensure the generated rationales satisfy the corresponding standard.
>
> **[A1]**: We agree that ensuring synthetic rationales align with ground-truth answers during the rationale generation stage is crucial in our method. In light of this, we use two approaches to validate the quality of the generated rationales:
>
> - **Substring match**: As introduced in **Section 2.2 (Line 185)**, we use a substring matching approach to assess their consistency on training samples. We find that the consistency ratio can reach 98% on average across five benchmarks, demonstrating the reliability of such synthetic rationales.
>
> - **LLM-as-a-judge**: In **Section 3.4 (Line 465)**, we also measure the quality of model-generated rationales using LLM-as-a-judge, which allows for a more comprehensive evaluation of the model output. Specifically, if the model-generated rationale is inaccurate despite the final answer being correct (probably due to the use of the LLM's parametric knowledge), the LLM judge will detect this inconsistency – please refer to **Appendix E** in our updated manuscript for a detailed example. \
> Given that LLM-as-a-judge can effectively assess the quality of the rationale, we also evaluate our method using GPT-4o as the judge. As shown in **Table 5(b)**, our method achieves even better performance under LLM based evaluation compared to pattern-matching based evaluation, which further confirms the reliability of the generated rationales.
>
> Therefore, for simplicity, we currently do not apply any additional enhancements to the synthetic rationales because this high consistency ratio and LLM-based evaluation already validate the quality of rationales, as also acknowledged by Reviewer MKpj.
>
> **[Q2]**: I have some questions about how to use rationale in evaluation. From what I understand, InstructRAG needs to generate rationales first, then concats the retrieved information and generated rationales to LM. Maybe the added computation cost needs to be analyzed in the paper.
>
> **[A2]**: We would like to clarify that our method does not involve a two-step inference process since rationales used in demonstrations are generated during the training process. As illustrated in **Figure 1**, our InstructRAG takes the same input as the standard RAG method (i.e., the question and retrieved passages). Therefore, there is no additional computation cost to use rationales in evaluation.
>
> Nonetheless, we acknowledge that our method might take slightly longer than standard RAG as it needs to generate more tokens (i.e., rationales) before concluding the answer. Per the reviewer’s suggestion, we conducted additional experiments on the PopQA dataset to compare the inference time of InstructRAG with standard RAG. To reduce randomness, we calculate the average inference time across 5 runs. The results show that our InstructRAG incurs negligible latency compared to standard RAG (approximately 0.05 seconds slower per sample) but delivers a significant improvement in generation accuracy (61.0% → 66.2%).
>
>
> Please let us know if you have any further questions, and we are happy to incorporate additional suggestions you might have! If you find our response satisfactory, we would be grateful if you could consider raising your score. Thanks again for your time and efforts!

---

> ### Author Response · Authors · 2024-12-02
> **A Gentle Reminder to Reviewer PDdA**
>
> Dear reviewer PDdA,
>
>
> We sincerely appreciate your valuable feedback. We would like to gently remind you that the discussion period will **end on December 2nd**.  In our rebuttal, we have provided comprehensive responses to your comments, and we hope our rebuttal satisfactorily addressed your concerns.
>
> Please let us know if you have any further questions, and we are happy to incorporate additional suggestions you might have. Thank you again for your time and efforts in reviewing our paper!

---

### Author Response · Authors · 2024-11-25
**A Gentle Reminder to Reviewers**

Dear Reviewers,

Thank you once again for your valuable time and constructive comments. In our rebuttal, we have provided thorough responses and additional results to address your concerns.

As the author-reviewer discussion deadline approaches, please let us know if our responses have satisfactorily addressed your questions. We would appreciate the opportunity to further address any remaining concerns you might have.

Best regards,

Submission#4836 Authors

---

### Meta-Review · Area_Chair_pvQL · 2024-12-15

**Metareview:**

This paper introduces InstructRAG, a novel framework for retrieval-augmented generation that leverages self-synthesized rationales to explicitly denoise retrieved content, enhancing both the interpretability and accuracy of generated answers. The method is applicable in both training-free and trainable settings, demonstrating consistent improvements over baselines across five knowledge-intensive benchmarks. Strengths include its minimal reliance on additional supervision, robust generalizability across tasks, and detailed experimental analysis. While some reviewers raised concerns about computational overhead and limited testing on diverse datasets, the rebuttal and additional experiments addressed these issues satisfactorily. This paper is recommended for Accept (spotlight) due to its strong contributions to enhancing RAG systems.

**Additional Comments On Reviewer Discussion:**

During the discussion, reviewers appreciated the paper's focus on explicit denoising and its methodological clarity. Concerns raised included the potential computational cost, generalizability across backbones, and applicability to noisy retrieval scenarios. The authors responded comprehensively, providing new experiments with diverse backbone models (e.g., LLaMA-3, Mistral, Qwen) and additional datasets (e.g., FreshQA). These additions strengthened the paper’s robustness and addressed concerns effectively, leading to score increases from multiple reviewers. Overall, the discussion reaffirmed the paper’s significant contributions and practical relevance.

---

### Decision · Program_Chairs · 2025-01-22

Accept (Poster)